# The Virtual Patient in Daily Orthodontics: Matching Intraoral and Facial Scans without Cone Beam Computed Tomography

Alessandra Campobasso [1,*], Giovanni Battista [1], Eleonora Lo Muzio [2] and Lorenzo Lo Muzio [1]

1 Department of Clinical and Experimental Medicine, University of Foggia, 71122 Foggia, Italy
2 Department of Translational Medicine and for Romagna, University of Ferrara, 44121 Ferrara, Italy
* Correspondence: alessandra.campobasso@unifg.it; Tel.: +39-0881-716576

**Abstract:** Aim: The authors provided an accurate, simple, and noninvasive method for matching the intraoral scan with facial scan of a patient, without the need of a cone beam computed tomography (CBCT). Materials and methods: Three different facial scans were acquired: the first one with the mouth closed, the second one with a voluntary "social smile", and the last one scanned the face with a lip-and-cheek retractor with dental arches in occlusion. The base of this method is to superimpose the area of the dental arches acquired by a face scanner with the same area derived by an intraoral scanner. Result: An accurate matching of intraoral and facial scans can be achieved without the risks of radiation exposure. Conclusions: The virtual patient helps the orthodontist to improve both diagnosis and treatment planning: a three-dimensional digital smile design can be performed, the patient's smile can then be analyzed in the context of the lips, and the teeth can be moved to achieve a consonant and balanced smile. All this information can be integrated in a clear aligner therapy or an indirect bonding procedure, enhancing outcomes in the facial esthetics.

**Keywords:** virtual patient; digital clone; intraoral scan; facial scan; digital smile design; digital orthodontics; cone beam computed tomography





## 1. Introduction

The analysis of facial and dental structures is essential for a correct orthodontic diagnosis and successful treatment planning. Conventional strategies for assessing dentofacial morphology are based on bi-dimensional (2D) imaging, obtained from photographs and 2D radiographs [1].

Currently, the latest digital technologies are rapidly changing orthodontics from a 2D to three-dimensional (3D) approach. The human face is a complex geometric structure, and it is difficult to realistically simulate the face only in a 2D image [2].

The development of 3D intraoral and facial scanners in the routine clinical practice has improved the diagnostic workflow and communication with patients [3–5]. Acquiring and integrating the 3D digital data provided by intraoral scans, facial scans, and cone beam computed tomography (CBCT), an orthodontist can easily create a "virtual patient" [6] or "digital clone" [1] in order to virtually plan an orthodontic treatment considering a 3D view of the patient [7].

With the improvements of optical technology, an intraoral scanner becomes fast, accurate, and comfortable for a patient. Moreover, current intraoral scanners do not require the once necessary antireflective powder, eliminating all the uncomfortable aspects of a conventional impression with the great advantage of an intraoral scanner to rescan select areas that may not have been adequately captured, without having to retake the entire impression [8]. The advancements in technology have not only improved the speed and accuracy of the intraoral scanners but also gained traction to the study of facial morphology using a three-dimensional face scanner [9,10].

Although intraoral and facial scanning has no detrimental effects on patients, clinicians frequently use the volumetric data obtained from CBCT to match the intraoral scans with

facial scans, exposing the patients to X-rays [1,3,6]. Actually, CBCT is not considered as a standard method for orthodontic diagnosis, and its routine clinical use is not acceptable as a substitute for conventional radiographs in both children and adults because of its higher radiation doses compared with 2D radiographs [10,11].

CBCT examinations should be performed only for valid diagnostic or treatment reasons and with the minimum exposure necessary for adequate image quality [12]: recommendations include retained and impacted permanent teeth, severe craniofacial anomalies, severe facial discrepancies with indication of orthodontic-surgical treatment, and bone irregularities or malformation of TMJ accompanied by signs and symptoms [13,14].

These recommendations are even more stronger in orthodontics and pediatric dentistry because children present tissues with higher radiosensitivity, greater number of cell divisions, and a longer lifespan for carcinogenesis development [15].

The aim of this paper was to provide a safe, X-ray-free, easy, and affordable method to combine the intraoral scan with facial scan of the patient without the need of cone beam computed tomography, and the accuracy of this technique is evaluated with a 3D reverse-modeling software package (Geomagic Control X—3D Systems, Rock Hill, SC, USA).

## 2. Technique

The process was demonstrated in a 25-year-old female patient.

1. A digital impression of both dental arches in occlusion was obtained with a TRIOS 3 color intraoral scanner (3Shape, Copenhagen, Denmark), and the data were exported as .stl (Standard Triangulation Language) or .ply (Polygon File Format) (Figure 1).

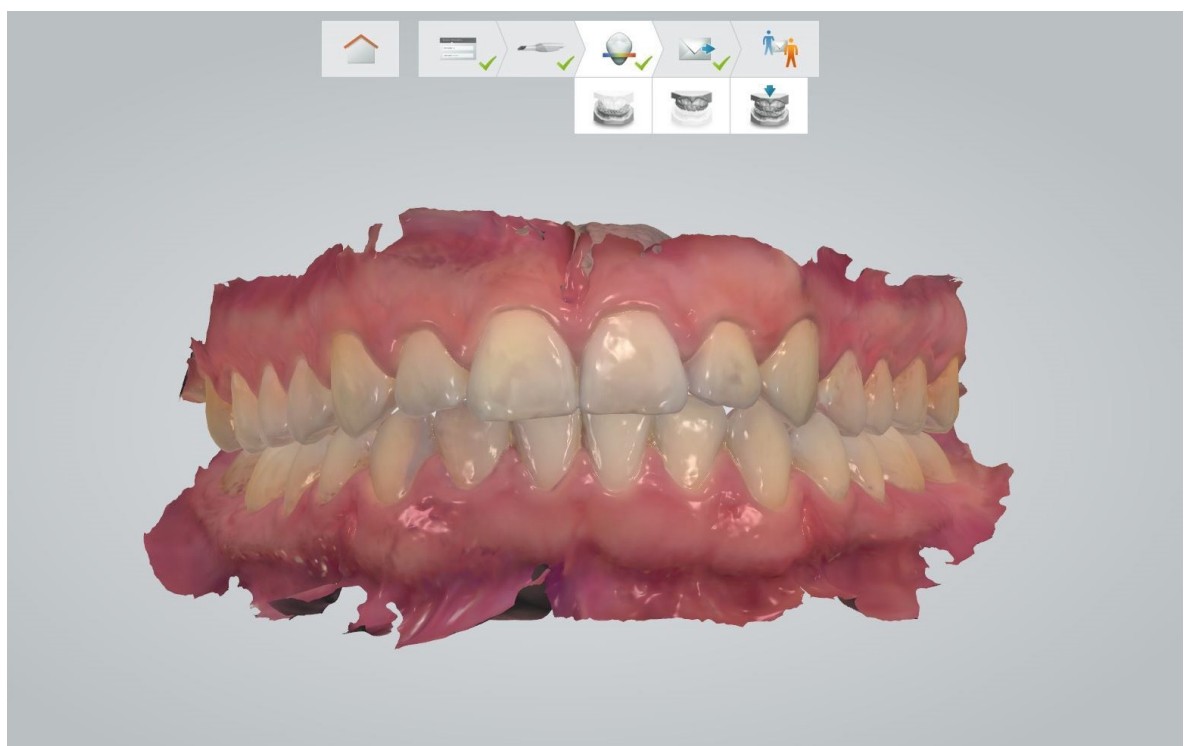

**Figure 1.** Digital impression of dental arches in occlusion with intraoral scanner.

2. A facial scan was acquired using a hybrid LED and infrared light source handheld color 3D scanner (EinScan H, Shining 3D Tech. Co. Ltd., Hangzhou, China). Based on the structured light technology of the LED and invisible infrared light, this scanner makes human face scanning more comfortable without strong light, even turning out to be safe for the patient's eyes [10].

Einscan H technical specifications show an accuracy of up to 0.05 mm and a depth of field of 200–700 mm (Table 1).

**Table 1.** Facial scanner technical specifications (Einscan H—Shining 3d—www.einscan.com, accessed on 7 September 2022).

| Light Source | White Light, Visible—Infrared Light, Invisible |
|---|---|
| Safety | LED light (eye-safe)—CLASS I (eye-safe) |
| Scan accuracy | Up to 0.05 mm |
| Volumetric accuracy | 0.05 + 0.1 mm/m |
| Scan and align speed | 1,200,000 points/s, 20 frames per second |
| Align modes | Markers alignment, feature alignment, hybrid alignment, texture alignment |
| Working distance | 470 mm |
| Depth of field | 200–1500 mm |
| Maximum FOV | 420 × 900 mm |
| Point distance | 0.25–3 mm |
| Color scanning | Yes |
| Output formats | OBJ; STL; PLY; P3; 3MF |
| Certifications | CE, FCC, ROHS, WEEE, KC |

The patient was seated on a chair 45–50 cm away from the scanner in a room with controlled illumination and was instructed to adopt the same facial expression and same position during the scan.

Three different facial scans were acquired using the standard mode setting: the first one in the natural head position with the mouth closed (Figure 2), the second one with a voluntary "social smile" (Figure 3), and the last one scanning the face with a lip-and-cheek retractor with dental arches in occlusion (Figure 4). Each facial scan was acquired in less than 10 s.

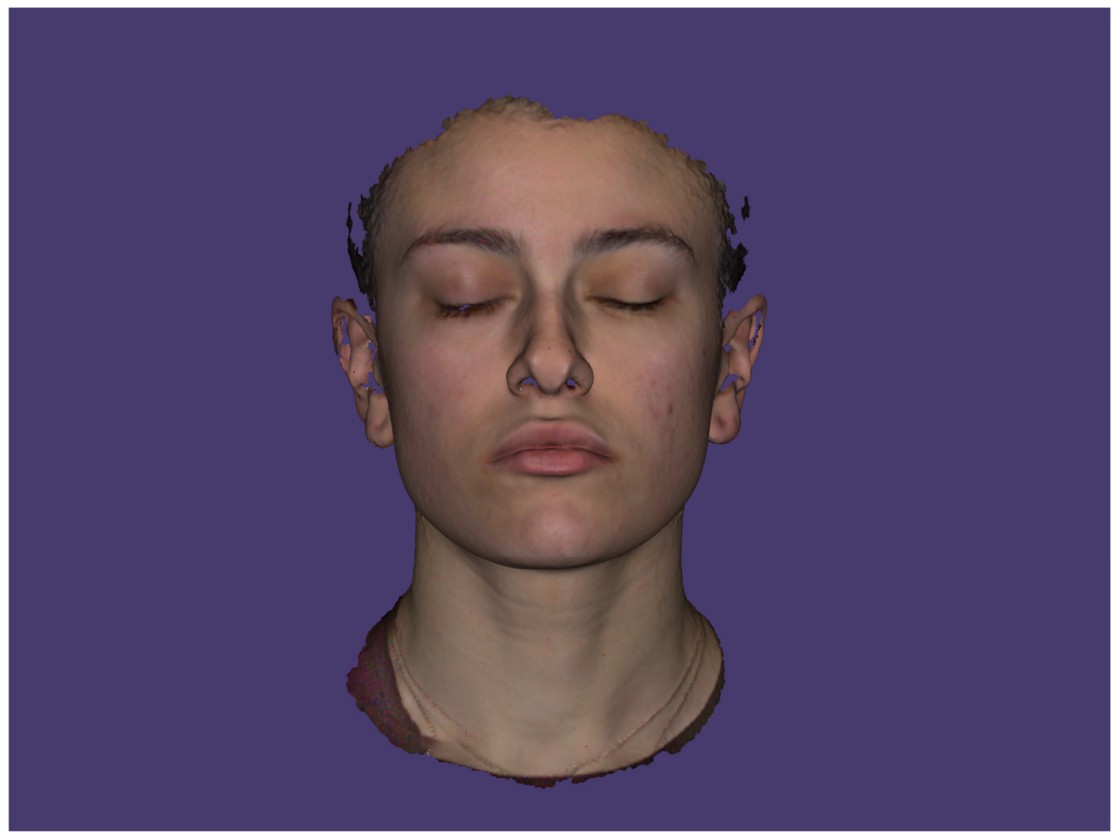

**Figure 2.** Face scan with closed mouth.

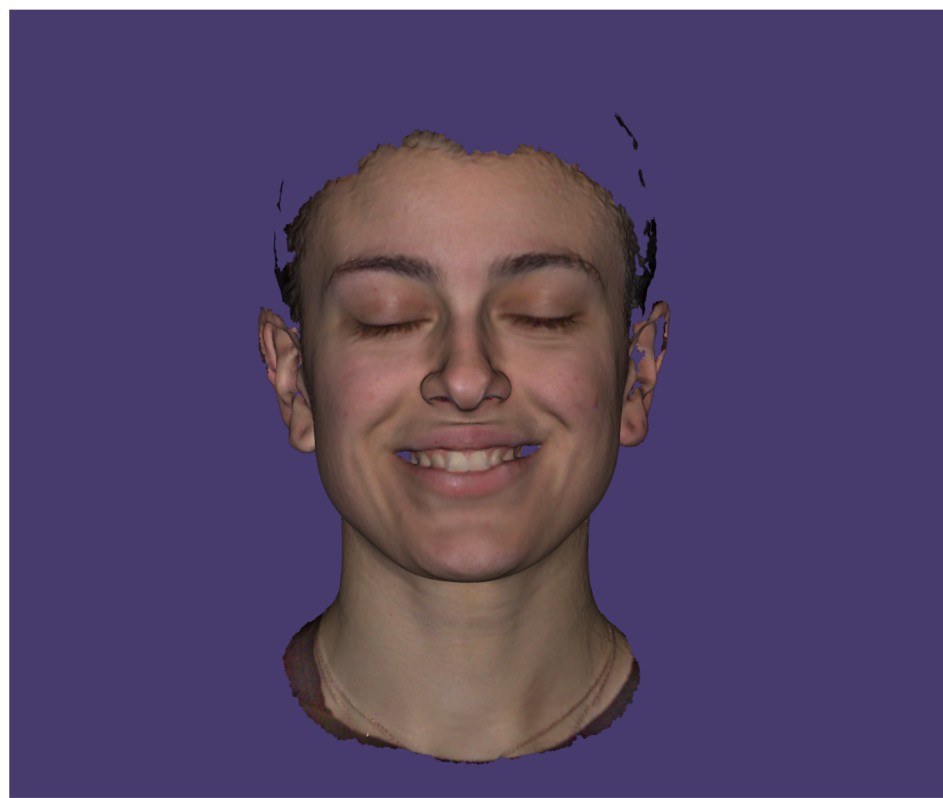

**Figure 3.** Face scan with voluntary "social smile".

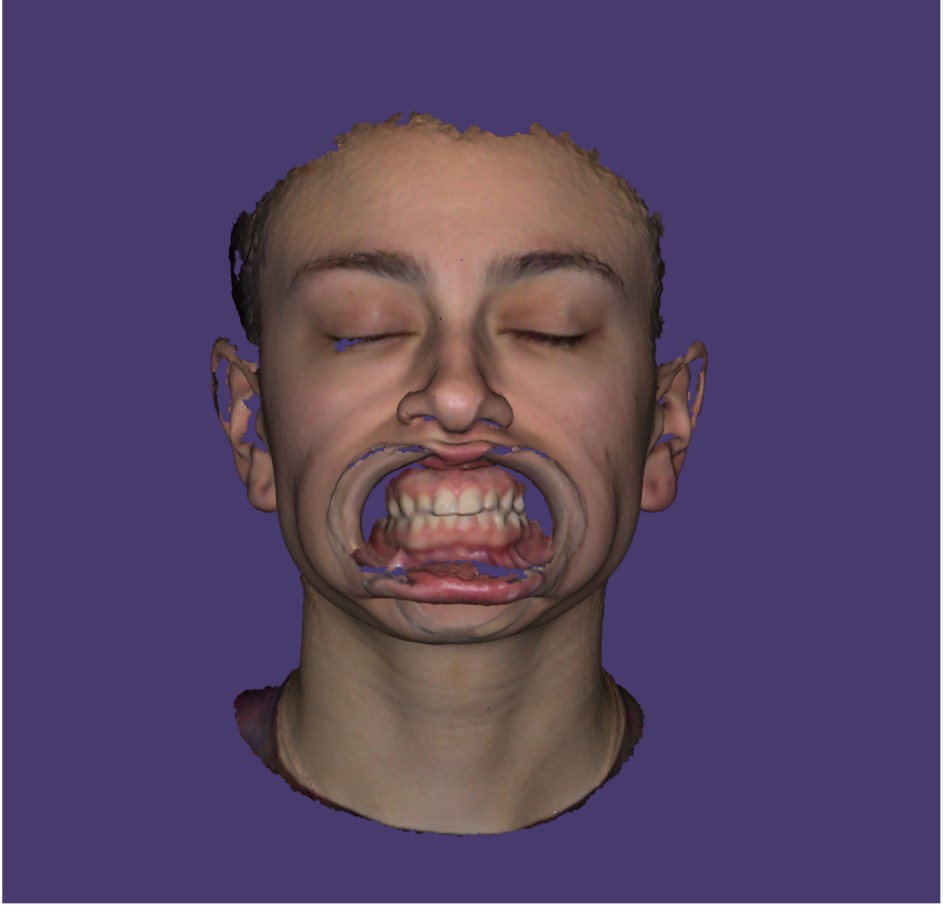

**Figure 4.** Face scan using a lip-and-cheek retractor.

These data were exported in a .stl or .ply file format.

3. Intraoral and facial scans were imported in Appliance Designer CAD software (3Shape, Copenhagen, Denmark) and matched together.

The face scan with a lip-and-cheek retractor was superimposed with the upper arch using the command "Align model, surface-3points", matching three dental points (the incisal edge of the upper incisor and the first upper premolars) (Figure 5).

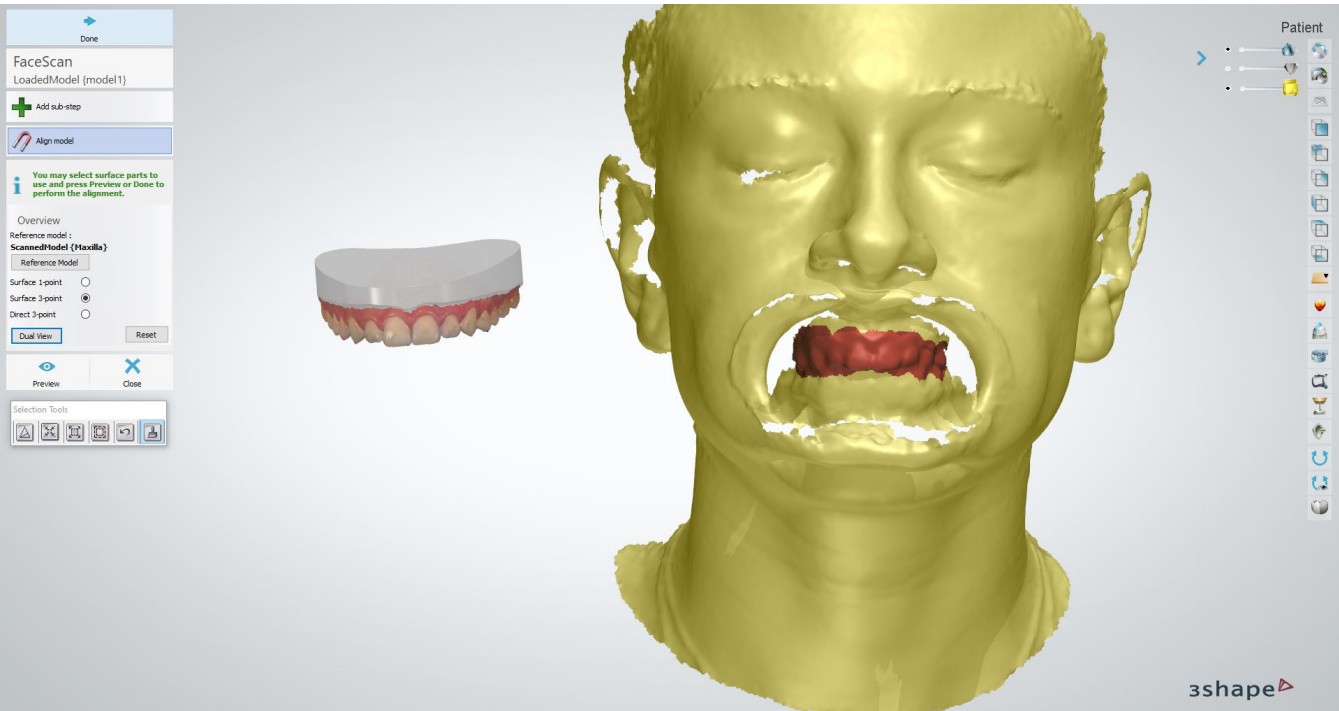

**Figure 5.** Matching the scan of the face with retractor with the upper arch.

This software performed one first matching between scans, aligning the corresponding three points of the tooth surfaces, then performed a fine superimposition using the "best-fit alignment" between the selected surfaces on the teeth.

Then, the face scan with the mouth closed and the scan with a smile were then matched with the scan with a cheek retractor using the same command "Align model-surface-3points", superimposing the surface of the front and the nose using 3 anatomical points (cutaneous glabella and endocanthion of the right and left eyes) (Figures 6 and 7).

4. The face scan with a social smile was then modified using the command "Modify model", and the portion of the dentition limited by the upper and lower lip line was removed (Figure 8).

5. It is also possible to superimpose the lateral cephalogram of the patient to the face and dental arches to combine the dental and esthetic features with the cephalometric analysis. Using the command "Cross-section", it is possible to overlay the cephalogram along a sagittal plane through the midline, aligning the picture with the anatomical references of the face and dentition of the patient (Figure 9).

The virtual patient was completed (Figures 10 and 11).

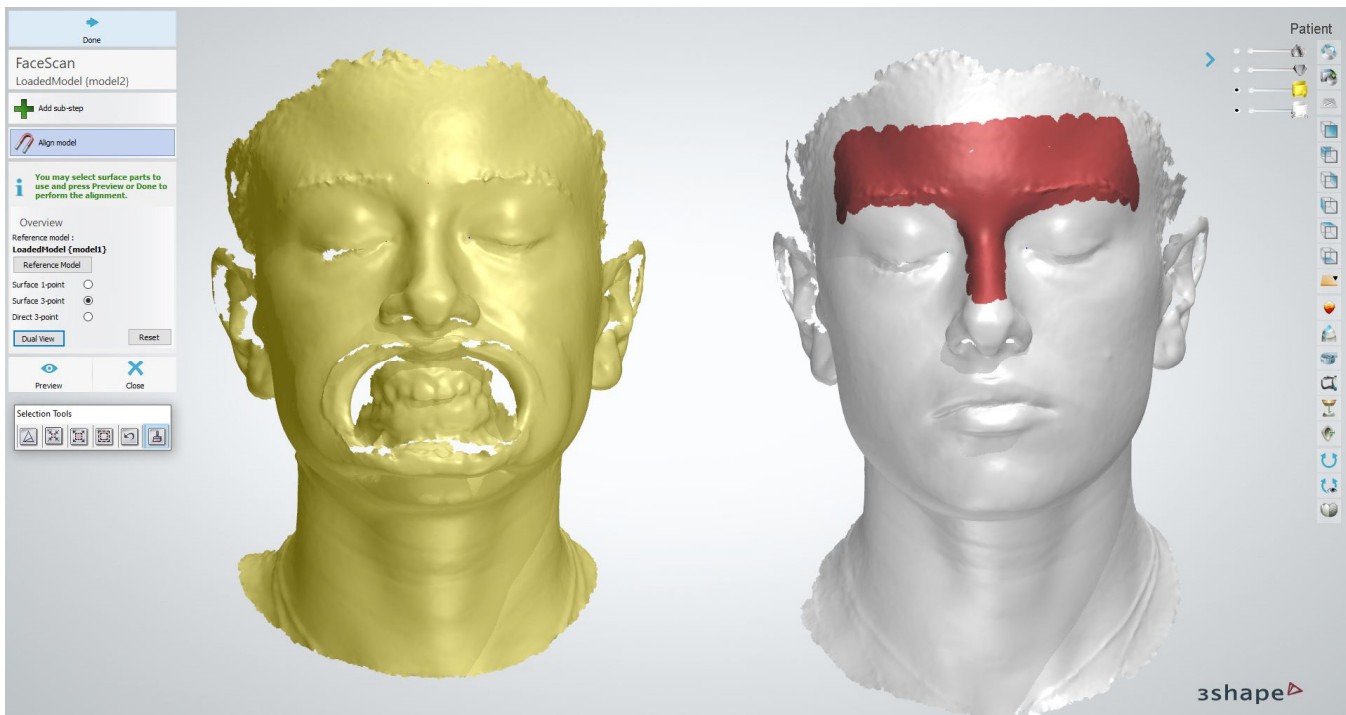

**Figure 6.** Matching the face scan with retractor with the face scan with closed mouth.

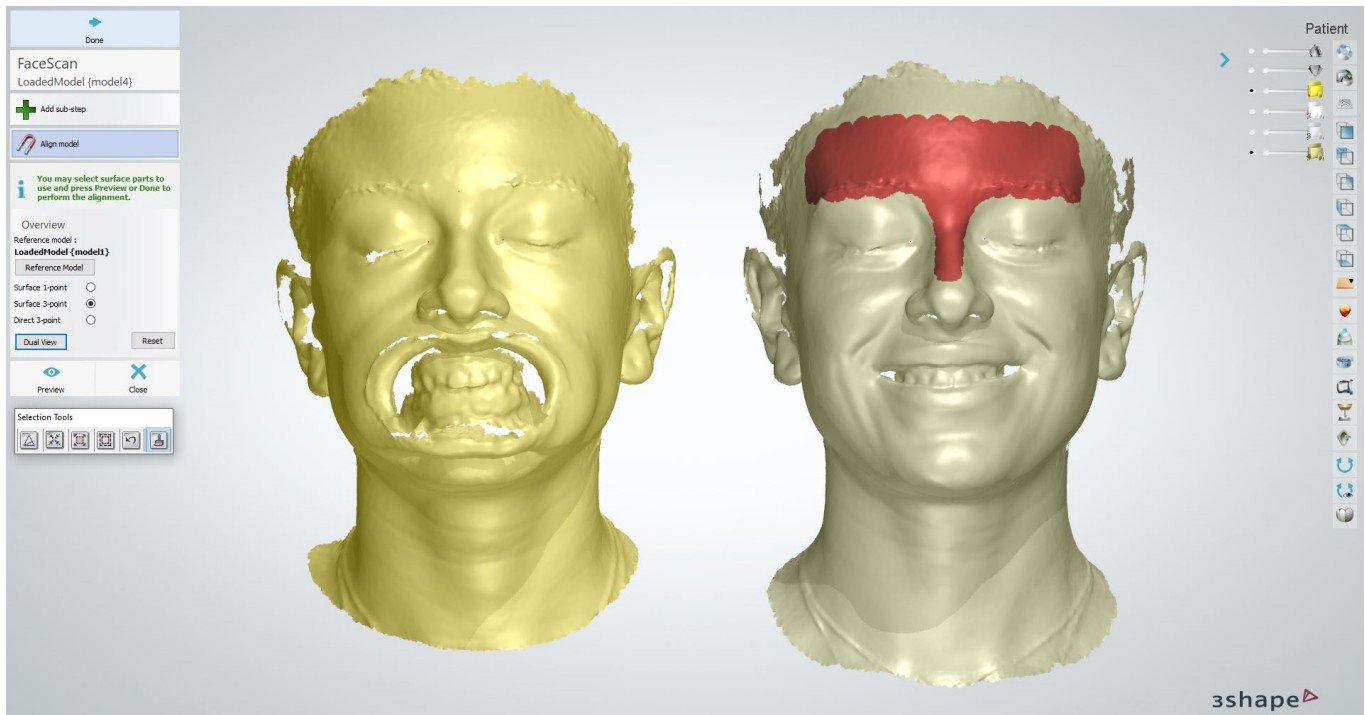

**Figure 7.** Matching the face scan with retractor with the face scan with smile.

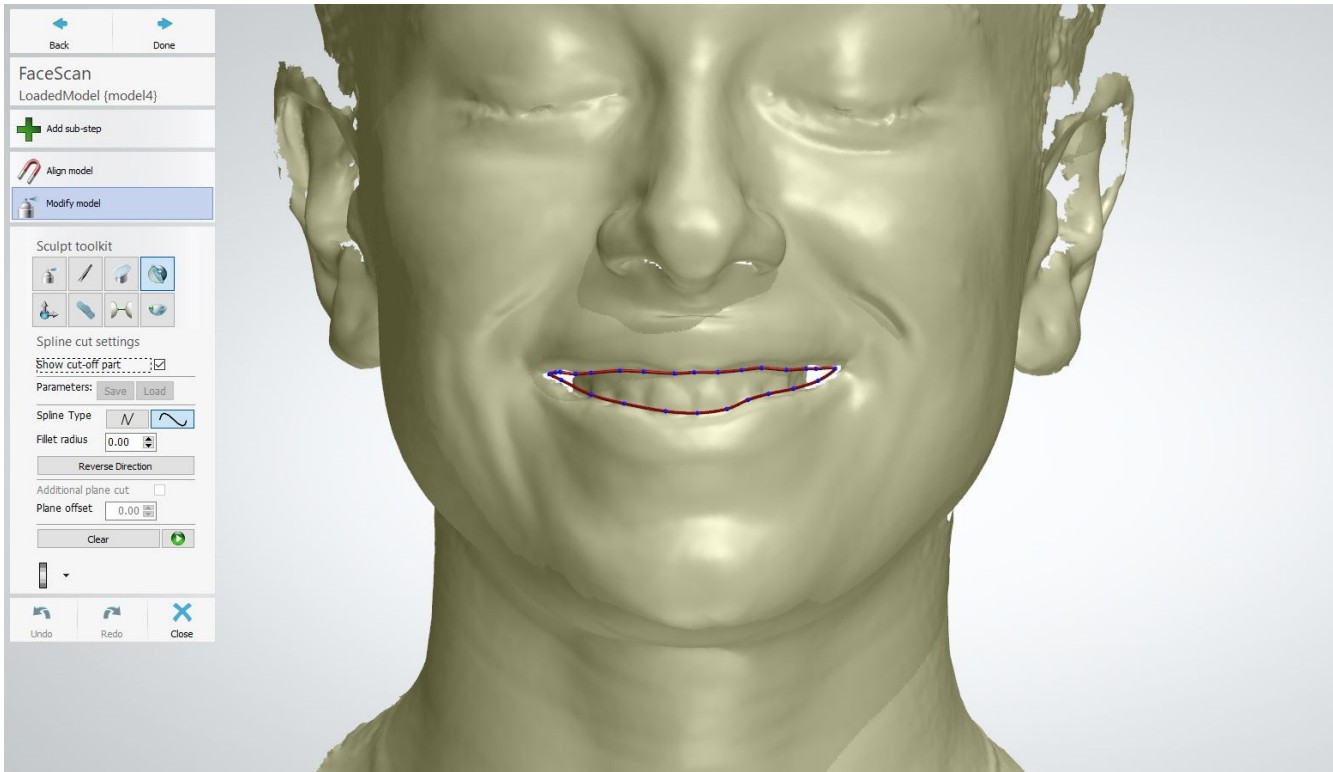

**Figure 8.** Upper arch is removed from face scan with the smile.

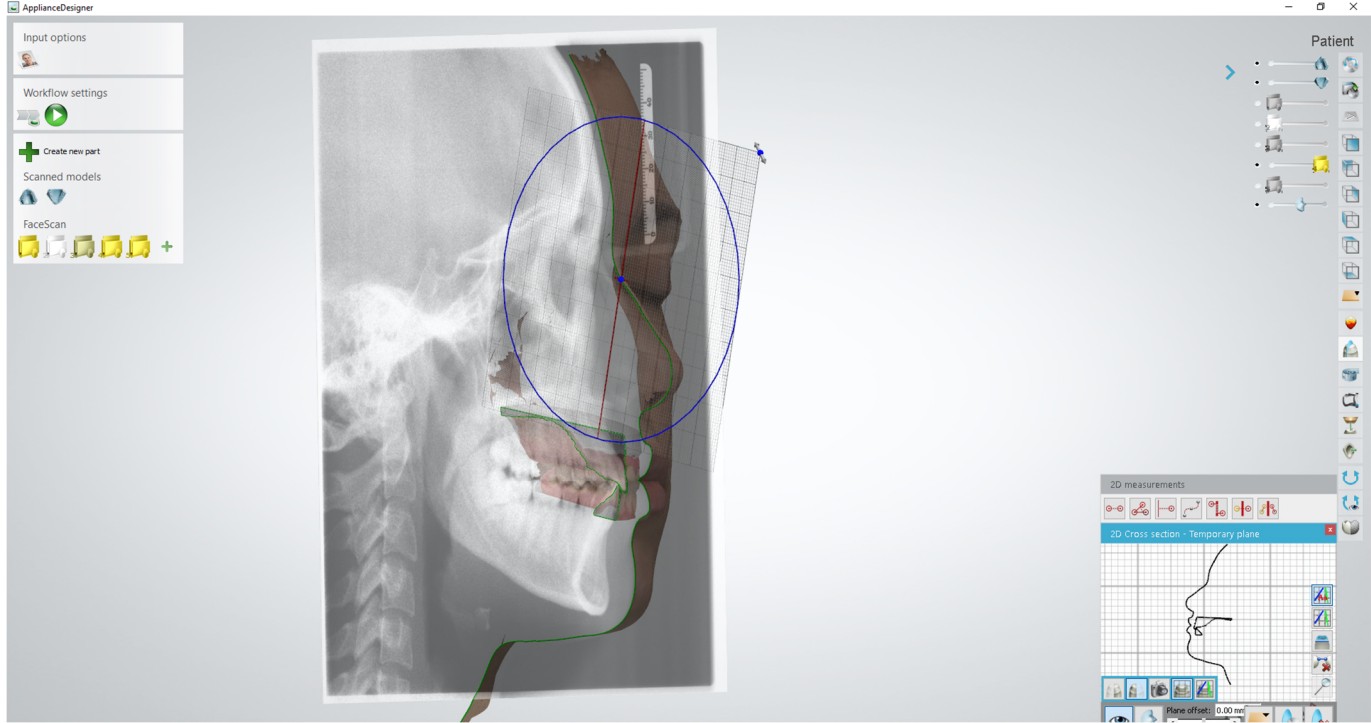

**Figure 9.** Lateral cephalogram is superimposed to the face scan.

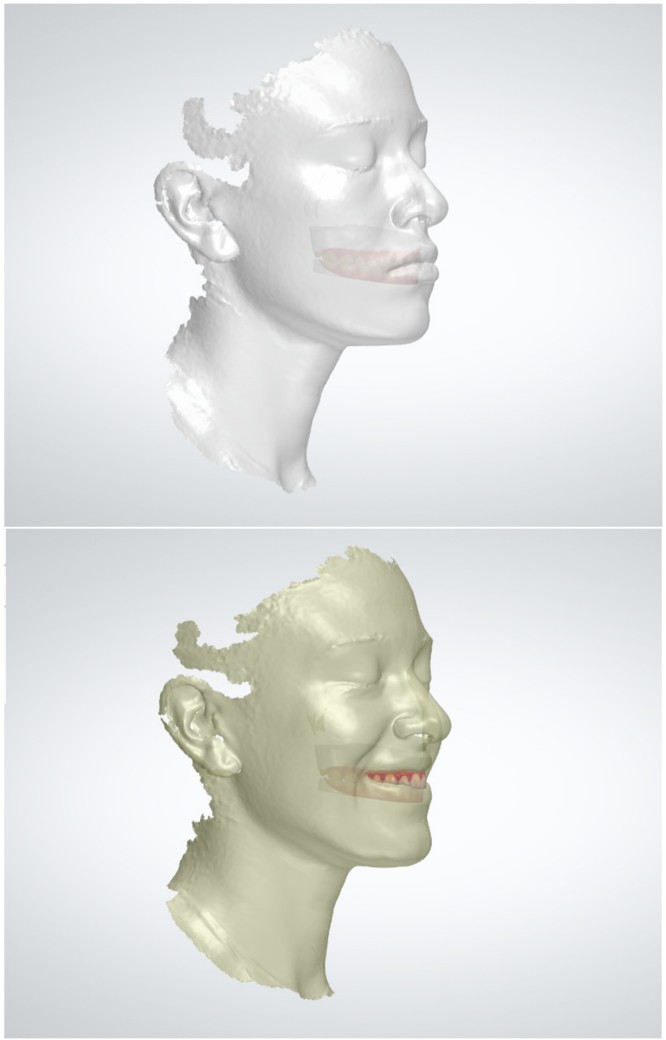

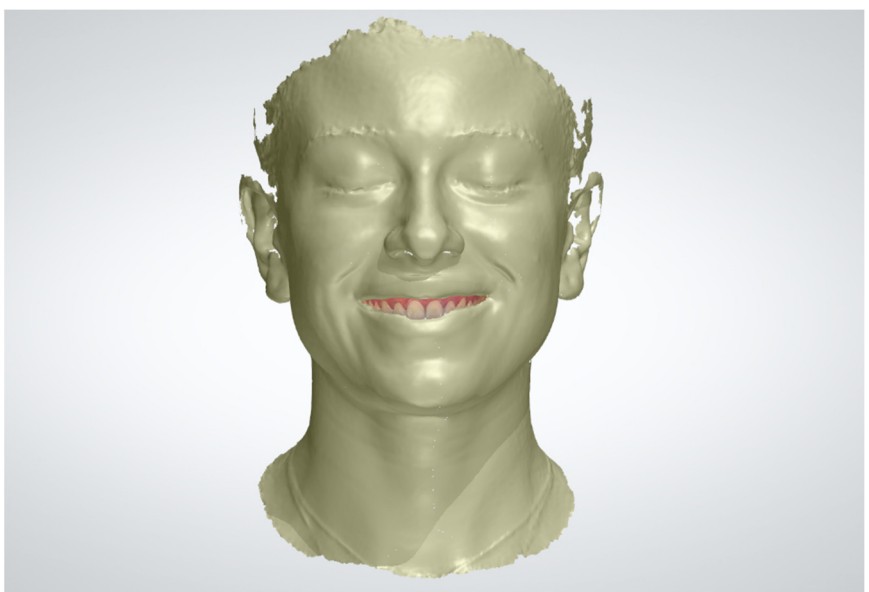

**Figure 10.** The virtual patient.

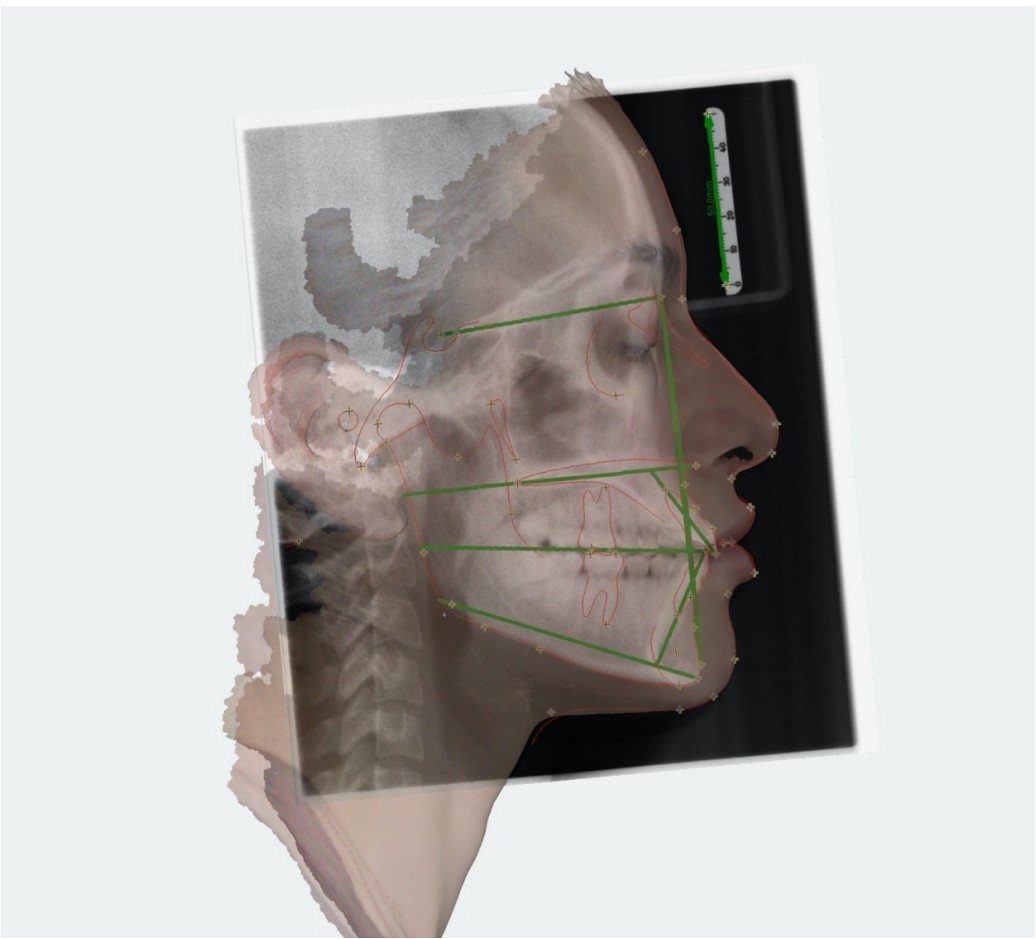

**Figure 11.** Face scan with virtual faceblow, position of condyles, and cephalometric landmarks.

The accuracy of this technique was then evaluated by exporting the intraoral and facial scans to a 3D reverse modeling software package (Geomagic Control X), calculating the deviation between surfaces.

At first, the distances between the corresponding areas of the teeth from the intraoral upper and facial scans with a lip-and-cheek retractor were compared to obtain color-coded maps. The tolerance range in green of the 3D deviation analysis was set to 0.1 mm with a maximum of 2.1058 mm. All the values in this range indicate the matching percentage between the intraoral and facial scans. The histogram chart shows a mean value of deviation of 0.2848 mm (Figure 12).

Subsequently, differences between the corresponding surfaces of the front and nose from the face scan with the mouth closed and from the face scan with a lip-and-cheek retractor were compared, and a color-coded map was obtained. The tolerance range in green of the 3D deviation analysis was set to 0.1 mm with a maximum of 4.4703 mm. The histogram chart shows a mean value of deviation of 0.2927 mm (Figure 13).

The last measurement was performed between the face scan with a smile and the face scan with a lip-and-cheek retractor, comparing the corresponding areas of the front and nose. The tolerance range in green of the 3D deviation analysis was set to 0.1 mm with a maximum of 4.4703 mm. The histogram chart shows a mean value of deviation of 0.4561 mm (Figure 14).

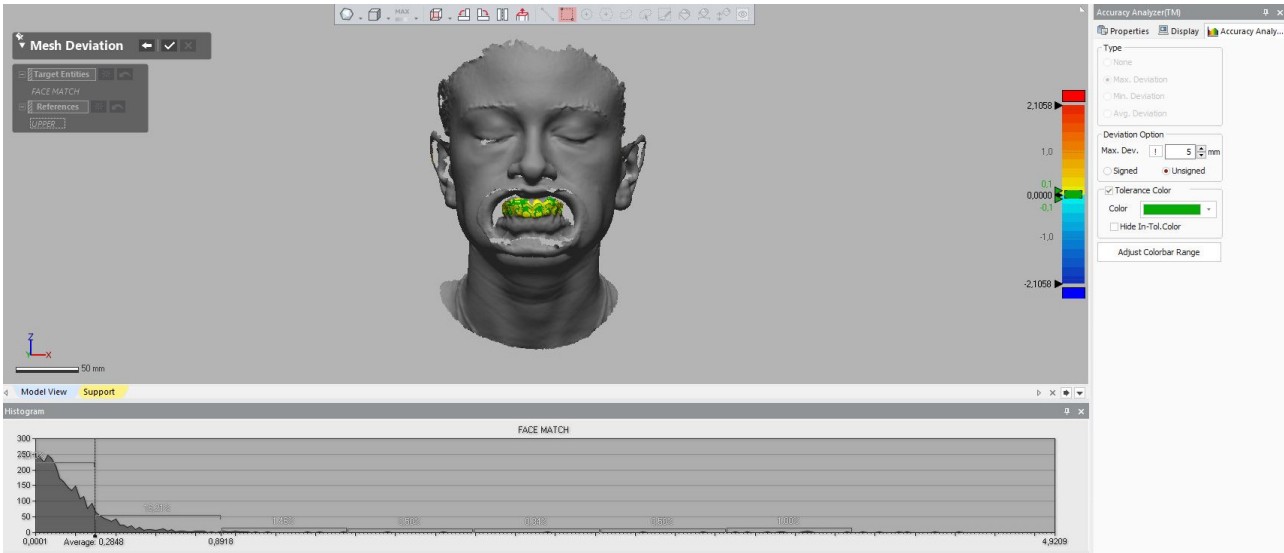

**Figure 12.** 3D deviation analysis between intraoral scan and face scan with lip-and-cheek retractor. RGB-colored scale bar (millimeters) is reported on the right. Histogram chart is reported at the bottom.

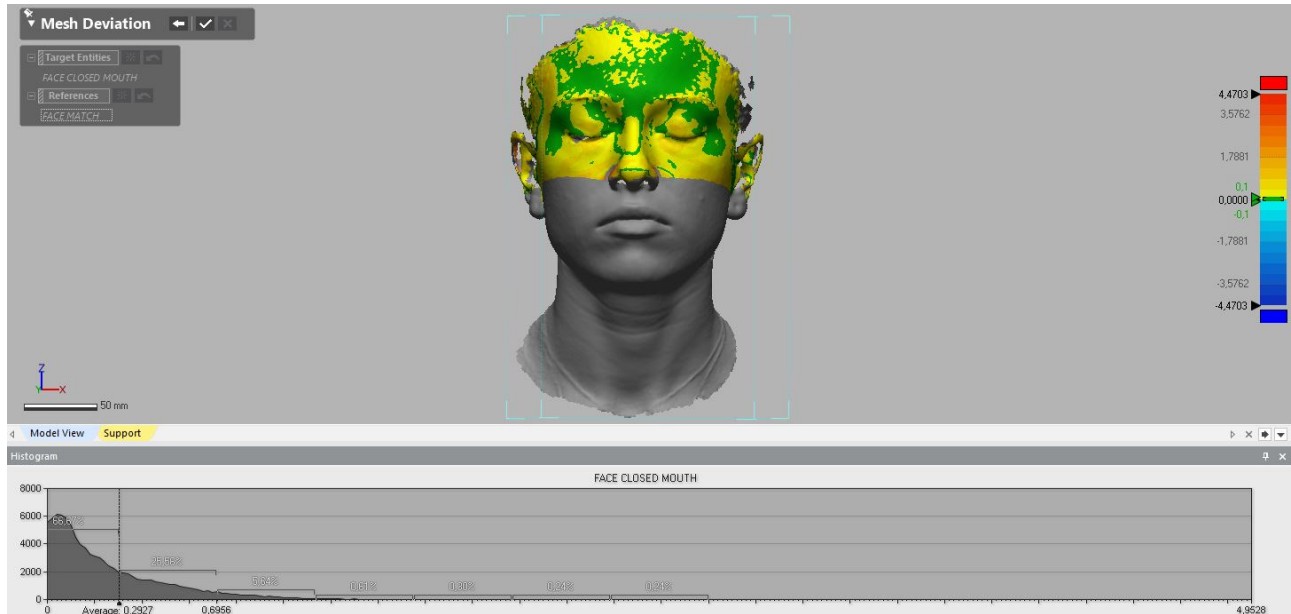

**Figure 13.** 3D deviation analysis between face scan with lip-and-cheek retractor and face scan with the mouth closed. RGB-colored scale bar (millimeters) is reported on the right. Histogram chart is reported at the bottom.

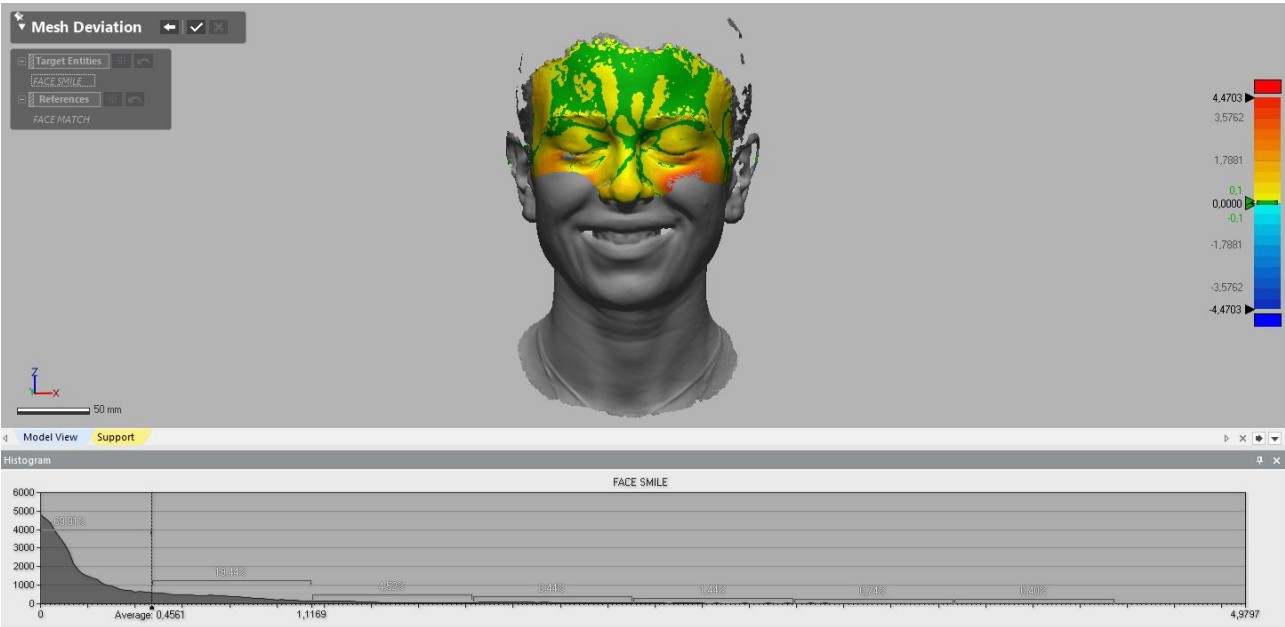

**Figure 14.** 3D deviation analysis between face scan with lip-and-cheek retractor and face scan with smile. RGB-colored scale bar (millimeters) is reported on the right. Histogram chart is reported at the bottom.

## 3. Discussion and Conclusions

The authors presented a simple technique to match the intraoral scans with the facial scans of the patient in an easy, affordable, and accurate way. This method can be used in daily orthodontic practice, and it is totally safe for patients.

The base of this technique is to superimpose the area of the dental arches acquired by a face scanner with the same area obtained by an intraoral scanner.

The accuracy of the face scanner is important: good achievement of this technique is correlated to the ability of the face scanner to provide sufficient precision when scanning tooth surfaces in comparison with an intraoral scanner.

Old-generation face scanners are used to scan a patient's head, but they need some external system of scan bodies to facilitate the superimposition of the face with the intraoral scan, resulting in a more difficult, complicated, and operator-dependent technique [16–20].

Another method consists of aligning the intraoral and facial scans using volumetric data from CBCT even if this exposes the patients to X-rays [1,6]. However, this method could not be performed if a CBCT is not available or necessary for orthodontic diagnosis, according to the concept "as low as diagnostically acceptable" (ALADA), especially in orthodontics and pediatric dentistry [12,13,21,22].

The accuracy of a CBCT is strictly related to the voxel dimension, FOV (field of view), and presence of soft tissue: high-resolution CBCT scans (voxel size 0.2 mm) have higher diagnostic accuracy than low-resolution CBCT (voxel size 0.5 mm) but expose the patient to a higher dose of X-ray [23]; a voxel size of 0.3 to 0.4 mm is adequate to provide CBCT images of acceptable diagnostic quality for treatment planning, but the voxel size and FOV must be taken into account to minimize patient radiation dose [24]; the soft tissue presence seems to affect the accuracy of the 3D hard tissue model obtained from a CBCT, below a generally accepted level of clinical significance of 1 mm [25].

Nowadays, the new generation of hybrid LED and infrared light source facial scanners based on structured light technology improves the accuracy and precision of the scan, making human face scanning more comfortable [9,26,27].

The Einscan H scanner has an accuracy of up to 0.05 mm, and future technological upgrades may still increase the precision and facilitate this technique.

However, to improve the acquired face scan quality, the authors suggest using a lip-and-cheek retractor in order to scan the dental surface as better as possible. It is also mandatory to acquire the tooth surfaces with a facial scanner at least from the upper right first premolar to upper left first premolar to perform a three-dimensional alignment between the facial and intraoral scans, and not only from the frontal perspective. This can resolve problems of misfit and error in the alignment of the occlusal plane, rotation, and palatal inclination of the teeth, which are very common in orthodontic cases.

Despite this facial scanner presenting a depth of field of 1500 mm, enough to scan the tooth surfaces from incisors to premolars and molars, in some patients with small opening of the mouth, the superimposition of the lateral cephalogram can be useful to align the intraoral scan to facial scan according to the occlusal plane [28].

This integration provides also information about the virtual faceblow, position of the condyles, and cephalometric landmarks [10] (Figure 11).

The accuracy of this technique was evaluated using 3D reverse modeling software Geomagic Control X. Deviations between scans were calculated and shown with a color-coded map. The tolerance range in green of the 3D deviation analysis was set to 0.1 mm for each superimposition, and the histogram chart showed the mean value of deviations.

The alignment between the intraoral and face scans with a lip-and-cheek retractor shows a mean value of deviation of 0.2848 mm: considering the lower accuracy of the facial scanner compared with the intraoral scanner, this result can be considered clinically acceptable [10] (Figure 12).

The alignment between the face scan with a "voluntary social smile" and the face scan with the mouth closed with the face scan with a lip-and-cheek retractor is simpler because of the wider surface of the front and nose and the easier identification of cutaneous landmarks. The results of the 3D deviation analysis showed a mean value lower than 0.5 mm, which can be considered clinically acceptable [10,27] (Figures 13 and 14).

Similar to the scanners, orthodontic software packages are also getting simpler and more user-friendly than those from the past years, helping clinicians to obtain a better task in less time.

In the future, coding some new orthodontic software packages or integrating the older ones with new specific workflow for face scanning might improve the efficiency and accuracy of this technique.

Different methods have been suggested to superimpose three-dimensional surfaces; these can be divided into two categories: landmark- and surface-based methods.

Landmark-based methods register 3D images on three or more manually selected corresponding anatomic landmarks. Surface-based registrations use anatomic areas as superimposition references, comparing the triangular representations of the corresponding 3D surface geometries [15].

Surface-based methods are based on "fine-matching": this technique reduces the errors associated with manually selecting superimposition landmarks, while the accuracy and reliability improve when 3D models are superimposed [29].

The software Appliance Designer allows matching the 3D images with the command "Align model-surface-3points" combining both the landmark and surface methods together with an increase in accuracy: one first alignment between scans is performed aligning the corresponding three landmarks, then a fine-matching using the "best-fit" algorithm between the selected surfaces on the different scans is performed.

The virtual dentofacial integration obtained by the present technique permits to create a virtual patient that can be evaluated in three dimensions.

This method can help the orthodontist improve the diagnosis and treatment planning: a 3D digital smile design can be performed, the patient's smile can then be analyzed in the context of the lips, and the teeth can be moved in relation to the curvature of the lower lip, intercommissural width, buccal corridors, and gingival architecture to achieve a consonant and balanced smile. All this information can be integrated in a clear aligner therapy or an indirect bonding procedure, enhancing outcomes in facial esthetics [30].

Furthermore, the growing interest in noninvasive diagnosis has allowed the development of new imaging tools without patients' exposure to ionizing radiation: a 3D cephalometric analysis can be performed starting from a facial scan, analyzing the facial morphology and soft tissue profile [31].

Patient monitoring can be also improved, comparing soft tissue changes during orthodontic, orthopedic, or surgical treatment, for a better communication with the patient [32,33].

The method proposed by the authors might be useful in daily clinical orthodontics, offering future possibilities also to design customized face-driven orthodontic appliances.

Further studies should be performed to investigate the accuracy of this technique in comparison with the standard method of aligning facial and intraoral scans with the CBCT data.

**Author Contributions:** Conceptualization, G.B.; writing—original draft preparation, A.C.; writing—review and editing, E.L.M.; supervision, L.L.M. All authors have read and agreed to the published version of the manuscript.

**Funding:** This research did not receive support from funding agencies in the public, commercial, or not-for-profit.

**Institutional Review Board Statement:** Not applicable.

**Informed Consent Statement:** Informed consent was obtained from the subject involved in the study. Written informed consent has been obtained from the patient to publish this paper.

**Data Availability Statement:** Not applicable.

**Conflicts of Interest:** The authors declare no conflict of interest.

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
