# Peer review of "The Virtual Patient in Daily Orthodontics: Matching Intraoral and Facial Scans without Cone Beam Computed Tomography"

_applsci, doi:10.3390/app12199870_

Round 1

Reviewer 1 Report

The key to this paper is whether facial scanners can accurately scan teeth. I can check the performance of the facial scanner through Figures 3 and 4. This scanner can never accurately attach teeth to face. Therefore, it is judged that this thesis was not written based on scientific evidence.

If you resubmit the more correct superimposition data, I will approve this paper. Thank you.

Author Response

Reviewer 1

The key to this paper is whether facial scanners can accurately scan teeth. I can check the performance of the facial scanner through Figures 3 and 4. This scanner can never accurately attach teeth to face. Therefore, it is judged that this thesis was not written based on scientific evidence.

If you resubmit the more correct superimposition data, I will approve this paper. Thank you.

We warmly thank the Reviewer for the suggestions that have significantly increased the quality of the manuscript.

Authors revised the manuscript adding the technical specification of the facial scanner (Table1) and provide new superimposition data.

The accuracy of this technique is evaluated using the 3D reverse modelling software Geomagic Control X and deviations between scans are calculated and shown with a colour-coded map. The tolerance range in green of the 3d deviation analysis was set to 0.1 millimeters for each superimposition and the histogram chart shows the mean value of deviations.

The alignment between intraoral scan and face scan with lip and cheek retractor shows a mean value of deviation of 0,0819 millimeters: considering the lower accuracy of the facial scanner (0.05millimeters) compared to the intraoral scanner, this result can be considerate clinically acceptable (Fig. 12).

The alignment between face scan with “voluntary social smile” and face scan with close mouth respectively with the face scan with lip and cheek retractor, presents a 3D deviation analysis with a mean value lower than 0.5 millimeters (Fig. 13-14).

These results (>0.5millimeters) are considerate clinically acceptable according to (Pellitteri F, Brucculeri L, Spedicato GA, Siciliani G, Lombardo L. Comparison of the accuracy of digital face scans obtained by two different scanners. Angle Orthod 2021;91(5):641-9) and (Antonacci D, Caponio VCA, Troiano G, Pompeo MG, Gianfreda F, Canullo L. Facial scanning technologies in the era of digital workflow: A systematic review and network meta-analysis. J Prosthodont Res 2022).

Reviewer 2 Report

The purpose of the technique is useful in everyday practice, but the authors must clarify:

-why is this technique more valuable than the one with photographs- the 3D face scanners are not usual devices in clinical practice and imply additional costs- Petre, A., Drafta, S., Stefanescu, C., & Oancea, L. (2019). Virtual facebow technique using standardized background images. The Journal of Prosthetic Dentistry121(5), 724-728.

-the alignment of the intraoral scans is made only from the frontal perspective -how do the authors solve the problems of cases with rotation and palatal inclination of teeth, very common in orthodontic cases

-information of "virtual facebow" is needed in orthodontic treatment planning -it seems that a lateral cephalogram ( and x-ray radiation) is still needed despite the use of 3D face scanning

-a reference cited in the study (Wampfler JJ, Gkantidis N. Superimposition of serial 3-dimensional facial photographs to assess changes over time: A 250 systematic review. Am J Orthod Dentofacial Orthop 2021) concludes that: "The limited available evidence suggests that surface-based registration performs superiorly to landmark-based registration. A small rectangular area on the forehead and an area including the middle part of the nose and the lower wall of the orbital foramen showed promising results, comparable to the anterior cranial base superimposition"- how the measurements made for planning the treatment could be reliable? Did the authors make any kind of validation of the proposed technique?

Author Response

Reviewer 2

The purpose of the technique is useful in everyday practice, but the authors must clarify:

-why is this technique more valuable than the one with photographs- the 3D face scanners are not usual devices in clinical practice and imply additional costs- Petre, A., Drafta, S., Stefanescu, C., & Oancea, L. (2019). Virtual facebow technique using standardized background images. The Journal of Prosthetic Dentistry121(5), 724-728.

We warmly thank the Reviewer for the suggestions that have significantly increased the quality of the manuscript.

The human face is a complex geometric structure, and it is difficult to realistically simulate the face only in a 2D image. This technique is more valuable than the one with photographs because it provides volumetric 3d data, capable to evaluate the facial morphology (Mai HN, Kim J, Choi YH, Lee DH. Accuracy of Portable Face-Scanning Devices for Obtaining Three-Dimensional Face Models: A Systematic Review and Meta-Analysis. Int J Environ Res Public Health 2020).

Although a face scanner is required, this method does not need a cbct, resulting safe for the patient and less expensive for the practice, considering that prices of handheld facial scanner are decresing and are cheaper than a cbct.

-the alignment of the intraoral scans is made only from the frontal perspective -how do the authors solve the problems of cases with rotation and palatal inclination of teeth, very common in orthodontic cases

The manuscript was fully revised as suggested:

“Authors suggest using a lip and cheek retractor, in order to scan the dental surface as better as possible; it is also mandatory to acquire the teeth surface with the facial scanner at least from upper right first premolar to upper left first premolar to perform a three-dimensional alignment between facial and intraoral scans, not only from the frontal perspective. This can resolve the problem of the misfit and error in the alignment of the occlusal plane, rotation and palatal inclination of the teeth, very common in orthodontic cases”.

-information of "virtual facebow" is needed in orthodontic treatment planning -it seems that a lateral cephalogram ( and x-ray radiation) is still needed despite the use of 3D face scanning

The manuscript was fully revised as suggested:

“Despite this facial scanner presents a dept of field of 1500 millimeters, enough to scan the teeth surfaces from incisors to premolars and molars, in some patients with small opening of the mouth, the superimposition of the lateral cephalogram can be useful to align the intraoral scan to facial scan according to the occlusal plane (Ghanai S, Marmulla R, Wiechnik J, Muhling J, Kotrikova B. Computer-assisted three-dimensional surgical planning: 3D virtual articulator: technical note. Int J Oral Maxillofac Surg 2010;39(1):75-82)”.

According to (Antonacci D, Caponio VCA, Troiano G, Pompeo MG, Gianfreda F, Canullo L. Facial scanning technologies in the era of digital workflow: A systematic review and network meta-analysis. J Prosthodont Res 2022) “An accuracy of less than 0.60 mm would allow face scan to be used as a digital facebow in everyday dentistry”, Authors evaluate the accuracy of this technique using the 3D reverse modelling software Geomagic Control X.

Deviations between scans were calculated and shown with a colour-coded map. The tolerance range in green of the 3d deviation analysis was set to 0.1 millimeters for each superimposition and the histogram chart showed the mean value of deviations.

The alignment between intraoral scan and face scan with lip and cheek retractor shows a mean value of deviation of 0,0819 millimeters (Fig. 12); the alignment between face scan with lip and cheek retractor and face scan with closed mouth shows a mean value of deviation of 0,0427 millimeters (Figure 13). These results can be considerate clinically acceptable, and this allow face scan to be used as a digital facebow.

Considering that a cephalogram is mandatory for a correct orthodontic treatment, specially in growing patient, the integration of facial and intraoral scans with lateral ceph could be considerate cost effective, because does not need a new x-ray exposure (cbct) for the patient.

Furthermore, lateral ceph exposes the patient to less radiation compared to cbct, resulting safer.

-a reference cited in the study (Wampfler JJ, Gkantidis N. Superimposition of serial 3-dimensional facial photographs to assess changes over time: A 250 systematic review. Am J Orthod Dentofacial Orthop 2021) concludes that: "The limited available evidence suggests that surface-based registration performs superiorly to landmark-based registration. A small rectangular area on the forehead and an area including the middle part of the nose and the lower wall of the orbital foramen showed promising results, comparable to the anterior cranial base superimposition"- how the measurements made for planning the treatment could be reliable? Did the authors make any kind of validation of the proposed technique?

Authors revised the manuscript adding the technical specification of the facial scanner (Table1) and provide new superimposition data measured by Geomagic Control X (Fig. 12-13-14).

According to (Pellitteri F, Brucculeri L, Spedicato GA, Siciliani G, Lombardo L. Comparison of the accuracy of digital face scans obtained by two different scanners. Angle Orthod 2021;91(5):641-9) structured light face scanners are accurate tool for clinical examination, with a margin of error between 0.5mm and -0.5mm. They concluded with: “Three-dimensional scans of the facial surface provide an excellent analytical tool for routine clinical evaluation”.

Reviewer 3 Report

The article presented is very interesting and proposes a method of morphological evaluation of the face, occlusion and cephalometric points that can be used in daily clinic to avoid the use of CBCT. The reading is quick and entertaining. I suggest to improve the article the following:
1.-Give more explicit information about the accuracy of the images that are created of the virtual patient.
2.-The superimposition of the cephalogram is only mentioned, but it would be interesting to be able to see an image that shows the craniometric reference points and proper angles of the cephalogram in the image of the virtual patient.

Author Response

Reviewer 3

The article presented is very interesting and proposes a method of morphological evaluation of the face, occlusion and cephalometric points that can be used in daily clinic to avoid the use of CBCT. The reading is quick and entertaining. I suggest to improve the article the following:
1.-Give more explicit information about the accuracy of the images that are created of the virtual patient.

We warmly thank the Reviewer for the suggestions that have significantly increased the quality of the manuscript.

Authors improved the manuscript as suggested, adding the technical specification of the facial scanner (Table1) and provide more information about the accuracy of the 3d images.

“The accuracy of the images was evaluated using the 3D reverse modelling software Geomagic Control X. Deviations between scans were calculated and shown with a colour-coded map. The tolerance range in green of the 3d deviation analysis was set to 0.1 millimeters for each superimposition and the histogram chart showed the mean value of deviations.

The alignment between intraoral scan and face scan with lip and cheek retractor shows a mean value of deviation of 0,0819 millimeters: considering the lower accuracy of the facial scanner (0.05millimeters) compared to the intraoral scanner, this result can be considerate clinically acceptable (Fig. 12).

The alignment between face scan with “voluntary social smile” and face scan with close mouth respectively with the face scan with lip and cheek retractor are simpler because of the wider surface of the front and the nose and the easier identification of cutaneous landmarks. 3D deviation analysis showed a mean value minor than 0.5 millimeters and can be considerate clinically acceptable (Pellitteri F, Brucculeri L, Spedicato GA, Siciliani G, Lombardo L. Comparison of the accuracy of digital face scans obtained by two different scanners. Angle Orthod 2021;91(5):641-9).

(Antonacci D, Caponio VCA, Troiano G, Pompeo MG, Gianfreda F, Canullo L. Facial scanning technologies in the era of digital workflow: A systematic review and network meta-analysis. J Prosthodont Res 2022)”.

2.-The superimposition of the cephalogram is only mentioned, but it would be interesting to be able to see an image that shows the craniometric reference points and proper angles of the cephalogram in the image of the virtual patient.

Author added new image showing the craniometric reference points and proper angles of the cephalogram in the image of the virtual patient (Fig. 11).

Reviewer 4 Report

1.       There is no novelty in the presented method of matching 3d scans of dentition and face.

2.       The method presented in the article is not widely used due to the fact that existing facial scanners cannot provide sufficient accuracy when scanning teeth. One of the reasons for this inaccuracy is due to the relatively lower resolution of scanning with a facial scanner in comparison with intraoral scanners. Another reason is caused by the use of a longer wavelength of structured light used for facial scanning, compared to intraoral dental scanning. Light from a longer wavelength penetrates through the transparent enamel, reflects off the surface of the teeth with errors, which sets an additional inaccuracy.

3.       To overcome the above problems, the South Korean company has created its own face scanner (http://www.raymedical.com/), which has a set of cameras for scanning the face and a set of cameras for simultaneous scanning of the dentition.

4.       The technical solution presented by the authors of the article, in which the superimposition the area of the dental arches acquired by a face scanner with the same area derived by intraoral scanner, will be more inaccurate in comparison with the solution presented by the South Korean company. The article does not provide any data characterizing the accuracy of matching scans according to the proposed methodology.

Author Response

Reviewer 4

  1. There is no novelty in the presented method of matching 3d scans of dentition and face.

We warmly thank the Reviewer for the suggestions that have significantly increased the quality of the manuscript.

Authors shows a method of matching intraoral scan and facial scan of the patient, acquired by new dual LED & infrared facial scanners based on structured light technology.

Old generation face scanners were used in the past to scan the patient’s head, but they needed some external systems of scan body to facilitate the superimposition of the face with the intraoral scan, resulting in a more difficult, time-consuming, complicated and operator-dependent technique.

The other common method to perform a superimposition between scans is using the cbct, exposing the patient to x-ray (considering voxel dimension and FOV for alignment).

The novelty of this paper is about this new method of superimposition, simple and accurate, without the need of cbct or other external scan body to perform the matching. The digital workflow is fully described and the accuracy of the method using the 3D reverse modelling software Geomagic Control X is added (fig. 12-13-14).

  1. The method presented in the article is not widely used due to the fact that existing facial scanners cannot provide sufficient accuracy when scanning teeth. One of the reasons for this inaccuracy is due to the relatively lower resolution of scanning with a facial scanner in comparison with intraoral scanners. Another reason is caused by the use of a longer wavelength of structured light used for facial scanning, compared to intraoral dental scanning. Light from a longer wavelength penetrates through the transparent enamel, reflects off the surface of the teeth with errors, which sets an additional inaccuracy.

Authors improved the manuscript as suggested, adding the technical specification of the facial scanner (Table1) and provide more information about the accuracy of the 3d images.

“The accuracy of the images was evaluated using the 3D reverse modelling software Geomagic Control X. Deviations between scans were calculated and shown with a colour-coded map. The tolerance range in green of the 3d deviation analysis was set to 0.1 millimeters for each superimposition and the histogram chart showed the mean value of deviations.

The alignment between intraoral scan and face scan with lip and cheek retractor shows a mean value of deviation of 0,0819 millimeters: considering the lower accuracy of the facial scanner (0.05millimeters) compared to the intraoral scanner, this result can be considerate clinically acceptable (Fig. 12).

The alignment between face scan with “voluntary social smile” and face scan with close mouth respectively with the face scan with lip and cheek retractor are simpler because of the wider surface of the front and the nose and the easier identification of cutaneous landmarks. 3D deviation analysis showed a mean value lower than 0.5 millimeters and can be considerate clinically acceptable (Antonacci D, Caponio VCA, Troiano G, Pompeo MG, Gianfreda F, Canullo L. Facial scanning technologies in the era of digital workflow: A systematic review and network meta-analysis. J Prosthodont Res 2022)”.

According to (Pellitteri F, Brucculeri L, Spedicato GA, Siciliani G, Lombardo L. Comparison of the accuracy of digital face scans obtained by two different scanners. Angle Orthod 2021;91(5):641-9) structured light face scanners are accurate tool for clinical examination, with a margin of error between 0.5mm and -0.5mm.

They concluded with: “Three-dimensional scans of the facial surface provide an excellent analytical tool for routine clinical evaluation”.

  1. To overcome the above problems, the South Korean company has created its own face scanner (http://www.raymedical.com/), which has a set of cameras for scanning the face and a set of cameras for simultaneous scanning of the dentition.

RAYFace scanner from South Korean company presented 9 image sensors (high resolution camera, RGB Camera and Depth Camera).

  1. The technical solution presented by the authors of the article, in which the superimposition the area of the dental arches acquired by a face scanner with the same area derived by intraoral scanner, will be more inaccurate in comparison with the solution presented by the South Korean company. The article does not provide any data characterizing the accuracy of matching scans according to the proposed methodology.

One of the Authors tried the South Korean scanner RAY Face in his practice last year (In Italy it’s branded and sold by Micerium). This scanner has infrared cameras (TrueDepth Camera) for dimensional scanning and RGB Camera for color detection, both for facial and dental tissues. These 3D scans are similar to those obtained with smartphone TrueDepth camera (Bellus3d with the iPhone).

According to (Pellitteri F, Brucculeri L, Spedicato GA, Siciliani G, Lombardo L. Comparison of the accuracy of digital face scans obtained by two different scanners. Angle Orthod 2021;91(5):641-9), structured light facial scanner (Face Hunter - Zirkonzahn, Italy) and the TrueDepth Camera scanner (Dental Pro – Bellus 3d) “reproduce 3D subjects with comparable, consistent, and reproducible measurements as compared with manual measurements, and it does not appear that one or the other of the measuring tools is systematically more accurate”.

Authors improved the manuscript as suggested, providing data characterizing the accuracy of the matching scans according to the proposed technique.

“The accuracy of the images was evaluated using the 3D reverse modelling software Geomagic Control X. Deviations between scans were calculated and shown with a colour-coded map. The tolerance range in green of the 3d deviation analysis was set to 0.1 millimeters for each superimposition and the histogram chart showed the mean value of deviations.

The alignment between intraoral scan and face scan with lip and cheek retractor shows a mean value of deviation of 0,0819 millimeters: considering the lower accuracy of the facial scanner (0.05millimeters) compared to the intraoral scanner, this result can be considerate clinically acceptable (Fig. 12).

The alignment between face scan with “voluntary social smile” and face scan with close mouth respectively with the face scan with lip and cheek retractor are simpler because of the wider surface of the front and the nose and the easier identification of cutaneous landmarks. 3D deviation analysis showed a mean value lower than 0.5 millimeters and can be considerate clinically acceptable (Antonacci D, Caponio VCA, Troiano G, Pompeo MG, Gianfreda F, Canullo L. Facial scanning technologies in the era of digital workflow: A systematic review and network meta-analysis. J Prosthodont Res 2022)”.

Round 2

Reviewer 1 Report

A few days ago, I already checked this paper as ‘reject’ 
Because, facial scan figure in paper was not good quality. (I observed this figure’s tooth is like 2D image (not 3D image), and in frankly, I didn’t believe in this figure or scanning data quality )
So, although this author has a additional work to accept article, but I still evaluate this paper is reject. 

Sorry for the inconvenience to you. 

Author Response

Thank you very much for your reply.

Reviewer 2 Report

The authors clarified the raised issues.

Author Response

Thank you very much for your kind reply.

Reviewer 4 Report

1. The authors state that the use of a lip and cheek retractor at the stage of facial scanning allows to abandon the use of CBCT or additional devices attached to the teeth during facial scanning.

Firstly, additional devices are used only because of insufficient accuracy of intraoral scans and facial scan matching.

Secondly, the article does not provide any evidence that the proposed method provides the same or, at least, close position of intraoral scans, in comparison with the method of using CT or the method of using additional devices mounted on the dentition (as Face Hunter method).

2. The authors presented 3D deviation analysis (Fig.12 ), using Geomagic Control X software. Fig.12 shows the result of the matching. Authors determined the average deviation of the alignment result as 0.0819 mm.

Firstly, colored scale on the fig.12 shows that the specified value of 0.0819 mm corresponds to the green color of the scale. But we can see on the Fig.12, that only a few small areas have green color. It means that almost all surface of the scans is disconnected much more then 0,08 mm. Otherwise, almost the entire surface should be painted green.

Geomagic Deviation Option – Sets options for deviation analysis.

Signed – Shows the deviation color map as positive and negative distances.

Unsigned – Shows the deviation color map as an absolute distance.

I wonder which option did you use?

Secondly, it is impossible to get reliable conclusions from only one example.

I would like to get stl of the matched models to have a chance to verify  the quality of matching.

3. No satisfactory response is provided

4. Convincing evidence in favor of the proposed method of matching, would be the same position of the intraoral scans, comparing other methods of matching. The content of the article does not provide enough arguments to conclude, that the proposed method can successfully replace previously known ones.

Author Response

Reviewer 4

  1. The authors state that the use of a lip and cheek retractor at the stage of facial scanning allows to abandon the use of CBCT or additional devices attached to the teeth during facial scanning.

Firstly, additional devices are used only because of insufficient accuracy of intraoral scans and facial scan matching.

Secondly, the article does not provide any evidence that the proposed method provides the same or, at least, close position of intraoral scans, in comparison with the method of using CT or the method of using additional devices mounted on the dentition (as Face Hunter method).

Thank you very much for your kind reply.

Author wrote these sentences:

“Old generation face scanners were used to scan the patient’s head, but they needed some external system of scan bodies to facilitate the superimposition of the face with the intraoral scan, resulting in a more difficult, complicated and operator-dependent technique[16-20]”.

“With the evolution of technologies, new generation of hybrid LED & infrared light source facial scanner based on structured light technology improves the accuracy and the precision of the face scan, making human face scanning more comfortable[9, 26, 27]. This Einscan H scanner has an accuracy up to 0.05 millimeters and the future technological will improve and increase precision”.

This technique was not possible in the past with the old generation scanner due to their lower accuracy, in the present we can instead obtain a clinically acceptable results (accuracy lower then 0.6millimeters - Antonacci, D., et al., Facial scanning technologies in the era of digital workflow: A systematic review and network meta-analysis. J Prosthodont Res, 2022.), and in the future technological improvement will facilitate this technique.

Further studies should be assessed to validate the accuracy of this method in comparison to CBCT or to system with additional devices mounted on the teeth.

Validation of this method is the focus of a new paper that the Author are working on.

  1. The authors presented 3D deviation analysis (Fig.12), using Geomagic Control X software. Fig.12 shows the result of the matching. Authors determined the average deviation of the alignment result as 0.0819 mm.

Firstly, colored scale on the fig.12 shows that the specified value of 0.0819 mm corresponds to the green color of the scale. But we can see on the Fig.12, that only a few small areas have green color. It means that almost all surface of the scans is disconnected much more then 0,08 mm. Otherwise, almost the entire surface should be painted green.

Geomagic Deviation Option – Sets options for deviation analysis.

Signed – Shows the deviation color map as positive and negative distances.

Unsigned – Shows the deviation color map as an absolute distance. 

I wonder which option did you use?

Secondly, it is impossible to get reliable conclusions from only one example.

I would like to get stl of the matched models to have a chance to verify  the quality of matching.

Figure 12 shows the Mesh Deviation between maxillary intraoral scan and Face scan with lip and cheek retractor in a colour-coded map.

The tolerance range in green of the 3d deviation analysis is set to ±0.1 millimeters with a maximum of 2,119 millimeters. All the values in this range indicate the matching percentage between the intraoral and facial scans. The histogram chart shows a mean value of deviation of 0,0819 millimeters and the range in percentage 50.81 %+33,47%= 84,28% between minimum deviation of -0,2466mm and maximum deviation 0,4104mm.

If the tolerance range in green was set to ±0.5 millimeter (limit to consider the results clinically acceptable) the entire surface is almost painted green (see the attached file).

Authors used the setting “Signed (Shows the deviation color map as positive and negative distances)” for the matching (see the attached file).

This paper shows a simple method for matching intraoral and facial scan of the patient without cbct but it needs a proper validation. Autor inserted in the limitation of the study: “Further studies should be performed to investigate the accuracy of this technique in comparison to the standard method of aligning facial and intraoral scans with the CBCT data”.

  1. No satisfactory responseis provided

RAYFace scanner from South Korean company presented 9 image sensors (high resolution camera, RGB Camera and Depth Camera). Authors reported the brochure of this product (see the attached file).

  1. Convincing evidence in favor of the proposed method of matching, would be the same position of the intraoral scans, comparing other methods of matching. The content of the article does not provide enough arguments to conclude, that the proposed method can successfully replace previously known ones.

Thank you very much for your suggestion.

Authors agree with the reviewer about the necessity of “further study to investigate the accuracy of this technique in comparison to the standard method of aligning facial and intraoral scans with the CBCT data”.

Comparing this method to the others, the accuracy of a CBCT should be considered, keeping in mind the voxel dimensions: high-resolution CBCT scans (voxel size 0,2 millimeters) has higher diagnostic accuracy than low-resolution CBCT (voxel size 0,5 millimeters) but expose the patient to a higher dose of x ray (Cetmili, H., M. Tassoker, and S. Sener, Comparison of cone-beam computed tomography with bitewing radiography for detection of periodontal bone loss and assessment of effects of different voxel resolutions: an in vitro study. Oral Radiol, 2019. 35(2): p. 177-183). A voxel size of 0.3 to 0.4 mm is adequate to provide CBCT images of acceptable diagnostic quality for treatment planning but voxel size and FOV must be considered to minimize patient radiation dose (Fokas, G., et al., Accuracy of linear measurements on CBCT images related to presurgical implant treatment planning: A systematic review. Clin Oral Implants Res, 2018. 29 Suppl 16: p. 393-415).

Exposing a young patient to hi-resolution CBCT only to minimize the error of the matching between scans could not be ethically and legally acceptable. The use of low dose CBCT could be acceptable but it could have an accuracy similar to this method (without cbct) considering the voxel dimension 0.4-0,5millimenters of low-res cbct.

Authors strongly agree with the reviewer about the necessity of “further study to investigate the accuracy of this technique in comparison to the standard method of aligning facial and intraoral scans with the CBCT data”.

Round 3

Reviewer 4 Report

1. You are providing incorrect information. According to the manufacturer Einscan H scanner has an accuracy up to 0.6 millimeters. I want to note that in reality the accuracy is always lower than indicated by the manufacturer. The manufacturer usually indicates the maximum possible result. You do not provide any reliable results.

2. One should use the Unsigned option to estimate the deviation between the surfaces.

The Signed option indicates the accuracy of the alignment itself, but not the accuracy of the coincidence of the surfaces.

3. It is very good that you are currently working on comparing different matching methods. At the end of such a study, it will be possible to publish an article describing the proposed method with an evidence base that the proposed method can effectively replace the known ones.

Author Response

Authors thanks the Reviewer and the Editor for your kind suggestions (see attached file).
